# Whole-Genome Resequencing−Based Qualitative Trait Locus Mapping Correlated *yellow* with the Mutant Color in Honeybees, *Apis cerana cerana*

**DOI:** 10.3390/ani14060862

**Published:** 2024-03-11

**Authors:** Shanshan Shao, Qiang Huang, Yalin Pei, Junyan Hu, Zilong Wang, Lizhen Zhang, Xujiang He, Xiaobo Wu, Weiyu Yan

**Affiliations:** 1Honeybee Research Institute, Jiangxi Agricultural University, Nanchang 330045, China; sssaaasss_cool@126.com (S.S.); qiang-huang@live.com (Q.H.); peiyalin2024@163.com (Y.P.); 18070071870@163.com (J.H.); wzlcqbb@126.com (Z.W.); zlzcg@126.com (L.Z.); hexujiang3@163.com (X.H.); wuxiaobo21@163.com (X.W.); 2Jiangxi Key Laboratory of Honeybee Biology and Beekeeping, Jiangxi Agricultural University, Zhimin Ave. 1101, Nanchang 330045, China; 3Institute for Quality & Safety and Standards of Agricultural Products Research, Jiangxi Academy of Agricultural Sciences, Nanlian Road 602, Nanchang 330200, China

**Keywords:** *Apis cerana cerana*, resequencing, *brown* mutation, *yellow*, RNA interference

## Abstract

**Simple Summary:**

This study examined the genetic basis of a mutation in cuticle color in the honeybee *Apis cerana cerana* using genome resequencing of wild−type and mutant drones produced by a single virgin queen. A candidate locus was identified by calculating the Euclidean distance between mutants and wild types at each SNP, performing Lowess regression to fit a curve to these data, and setting a threshold of the top 0.5% Euclidean distance for candidate region selection. From this, genes with synonymous substitutions became candidate genes. One of these genes, the *yellow* gene, had a 2 bp deletion causing a frameshift mutation. RT−qPCR of this gene was performed on RNA extracted from mutant and wild−type drones; gene expression was only significantly different between wild types and mutants at the *yellow* gene. Finally, RNA interference silencing of the *yellow* gene was used to reduce *yellow* gene expression in workers and putatively result in a lighter coloration. These results indicate that the *yellow* gene participated in the body pigmentation, and its defect was responsible for the *brown* mutation. It promotes the understanding of the molecular basis of body coloration in honeybees, enriching the molecular mechanisms underlying insect pigmentation.

**Abstract:**

The honeybee, *Apis cerana cerana* (*Ac*), is an important pollinator and has adapted to the local ecological environment with relevant coloration. The cuticle coloration of the *brown* (*br*) mutant is brown instead of black in wild−type individuals. Therefore, this study aimed to identify and characterize the gene responsible for the *br* mutation. Genome resequencing with allele segregation measurement using Euclidean distance followed by Lowess regression analysis revealed that the color locus linked to the mutation was located on chromosome 11. A 2−base deletion on exon 4 was identified in the *g7628* (*yellow*) gene after genome assembly and sequence cloning. In addition, the cuticle color of the abdomen of worker bees changed from black to brown when a defect was induced in the *yellow* gene using short interfering RNA (siRNA); however, the survival rate did not decrease significantly. These results indicate that the *yellow* gene participated in the body pigmentation, and its defect was responsible for the *br* mutation. This study promotes the understanding of the molecular basis of body coloration in honeybees, enriching the molecular mechanisms underlying insect pigmentation.

## 1. Introduction

The striking insect body color diversity is indispensable to their adaptation and survival [1]. In addition to ecological studies, the precise genes and molecular mechanisms underlying insect coloration and patterns have been analyzed [1,2], and relevant studies have increasingly focused on several insects, such as *Drosophila* [3,4,5], *Bombyx mori* [6,7], *Bicyclus anynana* and *Papilio xuthus* [8,9], *Harlequin Ladybird* [10], *Schistocerca gregaria* [11], *Timema chumash* [12,13], *Rhodnius prolixus* [14], and *Bombus melanopygus* [15]. Among the pigments for insect coloration, melanins are ubiquitous, producing shades from black to reddish−brown. Their synthesis involves various enzymes and their corresponding genes [16,17], such as *tyrosine hydroxylase* (*TH*), *yellow* (*Yellow−f/f2*), *Dopa decarboxylase* (*DDC*), *Laccase 2*, *N−acetyldopamine transferase* (*aaNAT*), *ebony*, *Tan*, and *aspartate 1−decarboxylase* (*ADC*).

Honeybees are important pollinators [18] and have adapted to environments with diverse body colorations and patterns, which are indispensable in the adaptation and survival of various insects [1]. The colors of the main body of honeybees are different degrees of black and yellow and are mediated by the black gene (*bl*) and a polygene series based on genetic rules [19,20]. However, the specific genes and molecular properties responsible for this coloration are unclear, except that *Amyellow-y* knock-out by CRISPR/Cas9 led to a melanization defect of the adult cuticle [21].

Physiologically, the cuticle pigmentation of honeybees begins at the pupal stage [22] and extends to the adult stage [23] and is regulated by hormones [24] and correlated with genes involved in cuticle development, such as *AmproPO* [25], *Amlac2* [26], *AmelTwdl1*, *AmelTwdl2* and *Ampxd* [27], and *AmBurs α* and *AmBurs β* [28].

The cuticle color of adult honeybee mutant *brown* (*br*) is brown, unlike the black color of the wild−type bees, *Apis cerana cerana* (*Ac*), providing a good biomaterial for studying the molecular basis of honeybee body coloration. In this study, using molecular mapping, candidate gene screening, and functional verification, the *br*−mutant gene was identified, the *br*−mutant locus was located on chromosome 11, and a defect in the *yellow* gene was found to be responsible for the *br* mutation, enriching the molecular basis of honeybee body coloration.

## 2. Materials and Methods

### 2.1. Honeybees

Wild-type and *br*−mutant drones used for genome resequencing, sequence cloning, and quantitative reverse transcription PCR (RT−qPCR) for the *g7628* (*yellow*) gene were the F1 drones of a single queen, which was cultivated in a colony in the apiary of Shanghai Suosheng Biotechnology in Shanghai, China, and in this population, the drones showed two colors, yellow and black. The virgin queen was also obtained from the same location and cultivated in a colony at the Honeybee Research Institute, Jiangxi Agricultural University, Nanchang, China (28.46° N, 115.49° E) to harvest the wild−type and *br*−mutant drones. The workers for RNA interference (RNAi) were obtained from three colonies with a strong population at the Honeybee Research Institute, of which the terga of the abdomen of workers were black. All honeybees were *A. cerana*.

### 2.2. The Rearing and Number Statistics of the Wild−Type and br−Mutant Drones

The virgin queen was introduced to the queenless hive with a queen limiter at the hive entrance and was tended by workers, fed royal jelly, and reared as a mature queen. After seven days of development, the queen was removed and treated with CO_2_ for 5 min thrice and returned to the colony. Seven days later, the colony was inspected to ensure that the queen had laid eggs and to estimate the time of emergence of the drones of offspring. The frame covered with capped drones was placed in a dark incubator (85% humidity, 34 °C), and the drones of offspring were observed, counted, and cleared daily until the end of drone emergence.

### 2.3. DNA Extraction, Library Preparation, and Genome Resequencing

DNA was extracted from the thoraxes derived from 15 wild−type drones (Black17, 18, 19, 21, 23, 24, 25, 26, 28, 39, 40, 41, 42, 43, 44) and 15 *br*−mutant drones (Brown77, 78, 79, 80, 81, 82, 84, 85, 86, 88, 90, 98, 99, 101, 106) using StarSpin Animal DNA Kit (GenStar, Beijing, China). All DNA samples were constructed as genomic DNA libraries and high-throughput sequenced by Novogene using an Illumina PE150 system (Illumina, San Diego, CA, USA).

### 2.4. Read Mapping, SNP Calling, and Annotation of Variants

Clean reads were obtained from the raw reads by removing the adapter sequence, low-quality, and unidentified reads. High−quality reads were obtained, with a quality of ≥95.88% of the bases for Q20 and ≥89.49% for Q30 (Appendix A). All clean reads were mapped to the reference genome of *Ac* (GCA_011100585.1) using BWA version 0.7.8−r455 [29] with the parameter: “mem −t 4 −k 32 −M”, and the genome was annotated through BLASTp against the NCBI non-redundant peptide database (NR), with parameters setting at E−value 1 × 10^−5^ [30]. PCR or optical duplicates were removed using SAMtools version 0.1.19−44428 cd [31] with the command: “rmdup”. The mapping rate for all the samples was ≥98.04% (Appendix A). The single nucleotide polymorphisms (SNPs) were called using Genome Analysis Toolkit (GATK) software (Version 3.8) [32] and filtered with QualByDepth (QD) < 4.0, FisherStrand (FS) > 60.0, or RMSMappingQuality (MQ) < 40.0 to obtain high−quality SNPs. SNP annotation was performed using ANNOVAR software version 2013Aug23 [33]. Next, the SNPs were categorized as occurring in exonic regions (including synonymous or nonsynonymous SNPs), intronic regions, splicing sites, upstream or downstream regions, or intergenic regions.

### 2.5. Euclidean Distance Calculation

Euclidean distance (ED) was calculated at each SNP location using the following equation [34]:(1)$$ED=\sqrt{{\left(A_{mut}-A_{wt}\right)}^{2}+{\left(C_{mut}-C_{wt}\right)}^{2}+{\left(G_{mut}-G_{wt}\right)}^{2}+{\left(T_{mut}-T_{wt}\right)}^{2}}$$
where *mut* and *wt* represent the pools for mutant and wild type, respectively, and the letters *A*, *T*, *C*, and *G* represent the frequency of their corresponding DNA nucleotide. The data were fitted using Lowess regression, and the ED measurement on top 0.5% ED was set as the threshold. To increase the effect of large ED measurements and decrease the effects of low ED measurements/noise, the ED power was raised to four, and the peak regions above the threshold were defined as candidate regions. The SNPs above 99.5% ED and within the identified region were filtered for nonsynonymous SNPs, with which the genes were the candidate genes.

### 2.6. The Candidate Genes Analysis

The cleaned reads of the 30 individuals obtained by resequencing were assembled using ABySS 4.8.3 [35] with a k−mer size of 96. The coding region sequences (CDS) of the candidate genes screened above were blasted in each assembled database using BioEdit (version 7.2.5) [36,37] to verify the genomic sequences of these genes. Next, the variants and allele frequencies of each gene in all individuals were counted. Genes with mutant allele frequencies = 1 in brown individuals and <0.1 in black individuals were used for further analysis. The expected amino acid sequences of the screened genes were blasted using BLASTp (protein–protein blast) in the NCBI database (https://www.ncbi.nlm.nih.gov, accessed on 17 April 2021), and genes with amino acid variants in polymorphic sites were excluded. The gene *yellow* with the variants on the conserved sites in the conserved domain was selected for subsequent analysis following the analysis of the conserved domains in NCBI.

### 2.7. Cloning and RT−qPCR of yellow

Total RNA was isolated from 21 newly emerged drones (11 wild-type and 10 *br*−mutant individuals) using the TransZol Up Plus RNA Kit (TransGen Biotech, Beijing, China). Reverse transcription was performed using the Prime Script™ RT Reagent Kit with gDNA Eraser (Takara, Dalian, China). The complementary DNAs (cDNAs) of two wild types and two *br*−mutant individuals were used as templates for the amplification of the CDS of *yellow*. The PCR was conducted using 2 × T5 Super PCR Mix (TSINGKE, Beijing, China) with the primer sets (Forward: ATGTTTCGCGAAACATTCGTTCTTCTCGTGAGTTTGG; Reverse: TCAATTATTCTCCCACCAAAGAGAA) under the following conditions: 98 °C for 2 min; 35 cycles of 98 °C for 20 s, 60 °C for 30 s and 72 °C for 20 s; and 72 °C for 2 min. The PCR products were cloned using the pClone007 Versatile Simple Vector (TSINGKE, Beijing, China) and TreliefTM 5α Chemically Competent Cells (TSINGKE, Beijing, China) and sequenced using an ABI PRISM 3730XL analyzer (ABI, Foster City, CA, USA). The cDNAs of the 11 wild types and 10 *br*−mutant individuals were used as the templates for the RT−qPCR of *yellow* with primer sets (Forward: CGTGAGTTTGGCGTATCTGG; Reverse: ACGCATTCTCCGGGATGTAT). The following cycling conditions were used: 50 °C for 2 min, 95 °C for 5 min, followed by 40 cycles of 95 °C for 15 s and 60 °C for 1 min. *Acrpl13a* (Forward: TGGCCATTTACTTGGTCGTT; Reverse: GAGCACGGAAATGAAATGGT) and *Acrpl32* (Forward: AGTAAATTAAAGAGAAACTGGCGTAAA; Reverse: TTAAAACTTCCAGTTCCTTGACATTAT) were used as reference genes for RT-qPCR. Statistical analyses were performed by the SPSS software (IBM SPS Statistics, Rel. 22.0.0.0, Armonk, NY, USA), where the quantitative data on genes were analyzed using a one-way ANOVA.

### 2.8. RNA Interference (RNAi)

A short interfering RNA (siRNA) sequence targeting *yellow* (SiYell) based on the CDS was designed and synthesized by GenePharma (Shanghai, China). Negative control siRNA (NC), which is widely used as a control and has no effect on gene expression in bees [38], was used in the control group. The siRNA sequences are listed in Table 1.

To determine the injected time and dosage of siRNA, 1 μL of SiYell (0.5 μg/μL or 1 μg/μL), NC (0.5 μg/μL or 1 μg/μL), or nuclease-free water were injected into each pupa with white, pink, brown, dark brown eyes at the dorsal abdomen of both drones and worker bees. It showed that the younger pupae were more sensitive to the reagents, and high dosage led to the failure of emergence or high mortality, while pupae injected at the late development stage, the body color of adults showed no evident changes. The adult body color of worker bees had an obvious change when the pupae with brown eyes were injected, while the drones with brown eyes injected failed to emerge though they showed lighter color at the late development than that injected with water. Meanwhile, the pupae injected with a high dosage of NC showed a proportion of nonspecific color changes in adults. Overall, the worker pupae with pink eyes were collected from the combs obtained from the three hives with the same genetic background and cultivated in an incubator at 34 °C and 85% relative humidity. On the fifth day of pupal development, pupae with dark brown eyes but without body pigmentation were used as samples for the injection for *yellow* interference. Approximately 0.5 μL of SiYell (0.75 μg/μL), NC (0.75 μg/μL), or nuclease-free water were injected into each pupa at the dorsal abdomen between the third and fourth segment, a total of 78, 53, and 57 pupae were injected, respectively. The injection depth of the microinjector was adjusted to concentrate the reagents in the cuticle. The injected pupae were maintained in an incubator (34 °C and 85% relative humidity).

To determine the interference effect of the siRNA, three pupae from each group were obtained after 68 h of cultivation as the effect of siRNA could last for 48−72 h depending on the situation [39], and the abdomens were dissected for total RNA isolation using the TransZol Up Plus RNA Kit (TransGen Biotech, Beijing, China). Reverse transcription was performed using the Prime Script™ RT Reagent Kit with gDNA Eraser (Takara, Dalian, China), and the cDNAs were used as the templates for the RT-qPCR of *yellow* with primer sets (Forward: CAATATCGGCGGCCTGAATT; Reverse: CGGGAAGAATCTGGAACTCG) using the following cycling conditions: 50 °C for 2 min, 95 °C for 5 min, followed by 40 cycles of 95 °C for 15 s and 60 °C for 1 min. *Acrpl13a*, *Acrpl32*, and *Acrps18* (the primer sets of *Acrpl13a* and *Acrpl32,* see Section 2.7; and the primer set of *Acrps18* is forward: GATTCCCGATTGGTTTTTGA, reverse: CCCAATAATGACGCAAACCT) were used as the reference genes, which were stable during the development [40]. Statistical analyses were performed by the SPSS software (IBM SPS Statistics, Rel. 22.0.0.0), where the quantitative data on genes were analyzed using a one−way ANOVA.

The number of bees that succeeded to emerge was counted to calculate the emergence rate. The adult bees were gathered, the color of whom was observed in the same view one day after emergence, when the interferential ones showed light color, but then the color tended to darken to a certain degree the next day, while the color of adults injected with water remained dark from the first day onward. The individuals with color lighter than that in the negative group were considered as ones with color variation. Statistical analyses were performed by the SPSS software (IBM SPS Statistics, Rel. 22.0.0.0) using a chi-square test with a post hoc test.

## 3. Results

### 3.1. Genetic Mapping of the br−Mutant Locus

Wild-type and *br*−mutant drones from a single queen showed black and brown cuticle coloration, respectively (Figure 1A). In addition, the number of *br*−mutant drones was close to the number of wild−type drones (Figure 1B).

The candidate region linked to the locus was identified using resequencing and ED. Lowess regression [41] was used to fit the ED data, and the Lowess curve showed a dominant peak on chromosome 11 with mini peaks on other chromosomes (Figure 2A). To reduce the effects of low ED noise and enhance that of large ED measurements, the ED power was raised to four, and a single distinct peak on chromosome 11 was observed on the Lowess curve (Figure 2B). To narrow down the candidate region, the ED measurement on the top 0.5% ED was set as the threshold, which was 1.074. The candidate region was 0.936 MB from position 242173 to position 1177784 on chromosome 11, containing 64 genes.

### 3.2. Candidate Gene Analysis

To determine the candidate genes for *br* mutation, SNPs with ED measurements above 99.5% and within the identified region by genetic mapping were used for annotation analysis. Eighteen candidate genes had nonsynonymous SNPs, and 12 had gene annotation descriptions (Table 2). 

After the sequencing analysis of the candidate genes, the gene, *yellow* (Gene ID: *g7628*), showed a 2−base deletion on the predicted exon 5 with an allele frequency of 1 in *br* mutants with an expected value of 0, whereas in the wild-type individuals, the allele frequency was 0.07 with an expected value of 0. The 2−base deletion of *g7628* was in the conserved major royal jelly protein (MRJP) domain, which might cause a frameshift mutation (Figure 3A).

The CDS of *g7628* was cloned using cDNAs from the *br*−mutant and wild-type individuals. Six exons were determined, which were one exon less than the expected CDS, and the 2−base deletion was confirmed on exon 4, where a frameshift and a premature TAG stop codon occurred (Figure 3B). The candidate gene, *g7628,* was defined as *yellow*, and its expression in the wild types was significantly higher than that in the *br* mutants based on RT-qPCR analysis (*p* < 0.05) (Figure 3C).

### 3.3. Functional Analysis of yellow Using RNAi

To investigate the possible correlation between the candidate gene, *yellow*, and *Ac* pigmentation, the expression of *yellow* was disrupted in worker bees by injecting siRNA into the abdomen of the pupae. Approximately 74.4−78.9% of the pupae emerged as adult honeybees with all treatments, and no significant difference was observed (*p* > 0.05) (Table 3). In addition, 77.6% of the survivors treated with SiYell showed color variation one day after emergence, which was significantly higher than that in the control groups (*p* < 0.01) (Table 4). The efficiency of the interference was estimated by determining the expression of the target gene 68 h after the injection [39]. The SiYell−treated honeybees showed significantly lower *yellow* expression levels than those treated with NC or water (Figure 4D). The pigmentation of the newly emerged honeybees was not significantly different among the individuals with different treatments; however, one day after emergence, the SiYell-treated individuals exhibited a brown cuticle coloration in the tergum of the abdomen, whereas those treated with NC or water exhibited a black coloration (Figure 4A,B). In addition, individuals with larger sizes treated with SiYell exhibited less color change (Figure 4C).

## 4. Discussion

In this study, whole−genome resequencing with allele segregation measurement using ED, followed by Lowess regression analysis, was used to identify the putative region associated with the color mutant locus, as the parental strain information was unknown [34].

The ratio of the wild-type to the *br*−mutant drones in the F1 offspring of a single queen was close to 1:1 (Figure 1B), suggesting that a major gene might be responsible for coloration in the *br* mutant, similar to the black gene *bl*, suppressing the yellow phenotype as a Mendelian recessive gene in *Apis mellifera* [19]. Based on the sequencing analysis of the candidate genes, *yellow* was screened with a 2−base deletion on exon 4, which disrupted the MRJP domain [42], and its function was validated using RNAi. The SiYell−treated workers, targeting *yellow*, exhibited brown cuticle coloration in the tergum of the abdomen, similar to the natural *br* mutants, instead of the black coloration in the wild type, demonstrating that *yellow* played a role in *Ac* pigmentation and its alteration was responsible for the *br* mutant.

Similar phenomena have been observed in other insects. The *yellow* gene is the ortholog of the *Apis mellifera* gene, *yellow−y*, and its knockdown by CRISPR/Cas9 induced an alteration in the melanization of the adult cuticle, showing a yellowish cuticle in the body and appendages of the G1−mutant drones [21]. In other insects, the *yellow-y* gene played an important role in the normal black pigment of larvae and adults in *Agrotis ipsilon* [43], adults in *Tenebrio molitor* [44], eggs, larvae, and adults in *Spodoptera litura* [45,46], chorion in *Aedes albopictus* [47], and hindwings in *Tribolium castaneum* [48]. In addition, the natural *yellow-y* mutations in *Bombyx mori* and *Drosophila melanogaster* also result in the yellow cuticles of the larvae or adults [49,50].

*Yellow-y* may be involved in dopa−, dopamine−, or both melanin synthesis pathways [51], and the messenger RNA (mRNA) expression pattern was consistent with the development of black-pigmented regions [6], which was also observed in this study, where *yellow* was highly expressed in the wild−type individuals compared to the low expression abundance in the *br* mutants (Figure 3C). In addition, the honeybee queen mates with multiple drones [52], resulting in offspring of different sizes, which induced different color variants after injection.

In addition to its roles in body pigmentation, *yellow−y* is also involved in other physiological processes, such as egg hatching [46], conferring of the morphology of the outer endochorion [47], segmentation and larval molting [45], and male mating behaviors [53]. Given that the *br* mutant was rare in *Ac*, who have adapted with black coloration, and the yellow morphs have been shown to process significantly more nectar gatherers than the black morphs of *Apis cerana Fab.* [54], further investigation of the *yellow* gene function regarding physiological processes is required. Furthermore, *yellow* was not fatal according to the ratio between the *br*−mutant and wild−type drones in the natural colony and RNAi verification, suggesting that *yellow* is a potential genetic marker in breeding owing to its striking color in *br* mutants.

## 5. Conclusions

The color locus for the *br* mutant was located on chromosome 11 of *Ac*. The *yellow* gene was identified and confirmed to be involved in body color pigmentation, and its alteration was considered responsible for the *br* mutant. These results promote the understanding of the molecular basis of body coloration in honeybees.

## Figures and Tables

**Figure 1 animals-14-00862-f001:**
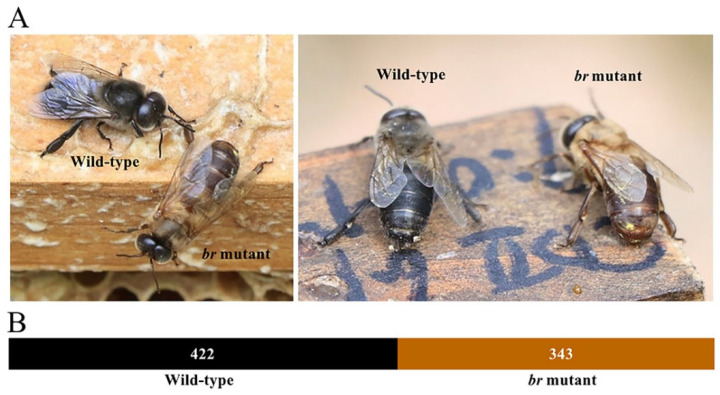
The phenotypes and number statistics of the wild−type and br−mutant drones, *Apis cerana cerana*. (**A**) The wild−type and *br*−mutant drones photographed under natural light. (**B**) The number of wild−type drones and *br*−mutant drones from one single queen.

**Figure 2 animals-14-00862-f002:**
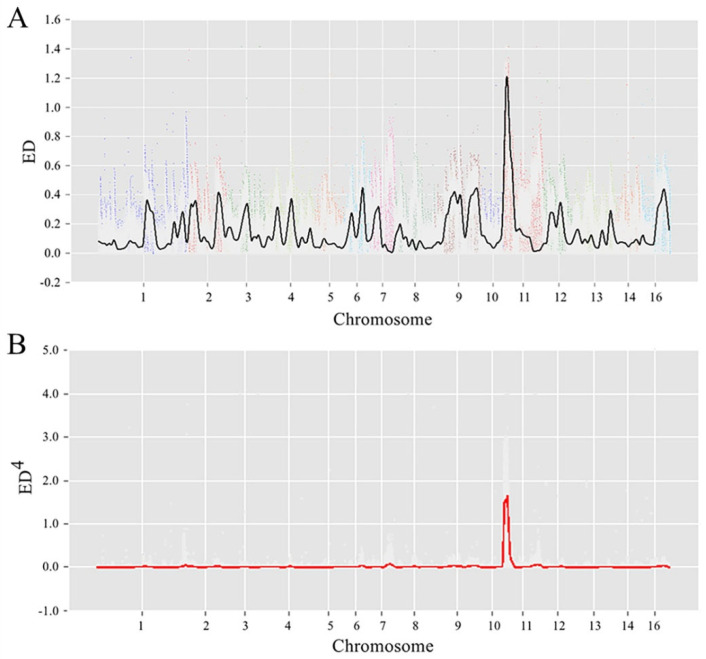
Genetic mapping of the *br*−mutant locus. (**A**) Raw Euclidean distance scores with the Lowess fit curve across the genome−wide. (**B**) Four powers of the Euclidean distance scores with the Lowess fit curve across the genome−wide.

**Figure 3 animals-14-00862-f003:**
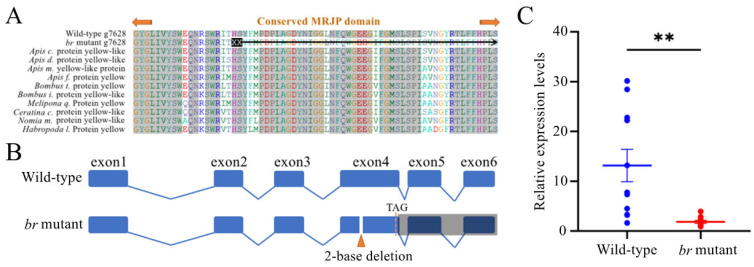
The analysis of the candidate gene, *g7628*. (**A**) The alignment of the expected amino acid sequences of *g7628* with the sequences of other species. (**B**) The comparison of *g7628* among wild types and *br* mutants. The *br* mutant had a 2−base deletion in exon 4 with a premature TAG stop codon, and the exons covered with shadow were untranslated regions. (**C**) The relative expression levels between wild types and *br* mutants. ** *p* < 0.01.

**Figure 4 animals-14-00862-f004:**
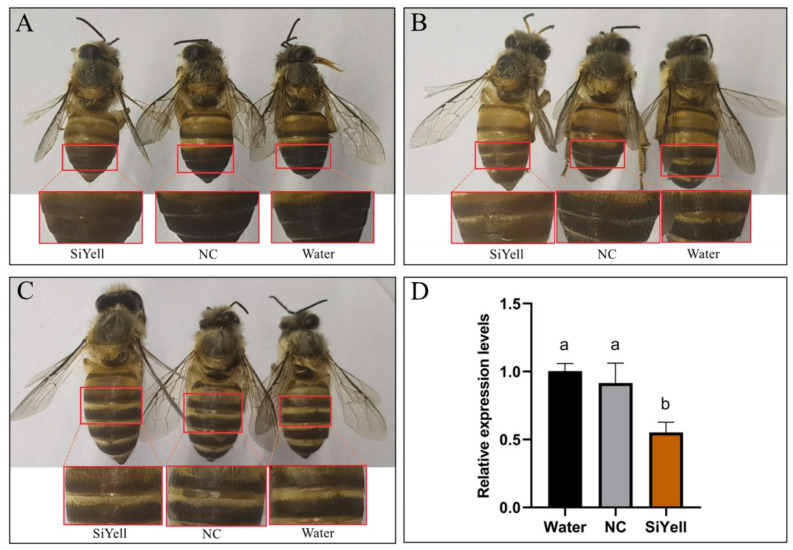
The effects of RNAi on the color phenotype and the expression of the target gene. (**A**–**C**) The color phenotypes of the one−day emerged workers after the treatment with SiYell, NC, or water. The regions inside the box were shown below the respective individual. (**D**) The expression of *yellow* at 68 h after the injection. Different letters above bars indicate significant difference.

**Table 1 animals-14-00862-t001:** SiRNA sequences for RNA interference.

siRNA	Sequence (5′→3′)	Chemical Modification
SiYell	sense: GCAUGAGCCUGUCGCCCAUTT	2′−Ome
	antisense: AUGGGCGACAGGCUCAUGCTT	
NC	sense: UUCUCCGAACGUGUCACGUTT	2′−Ome
	antisense: ACGUGACACGUUCGGAGAATT	

**Table 2 animals-14-00862-t002:** The description of the candidate genes identified with nonsynonymous SNPs.

Chromosome	Position	Gene ID	Gene Name	Gene Annotation
LG11	284475	g7587	PTPM1	Phosphatidylglycerophosphatase and protein-tyrosine phosphatase 1
LG11	310954	g7591	DNMT1	DNA (cytosine-5)−methyltransferase 1
LG11	453597	g7609	DGKH	Diacylglycerol kinase eta
LG11	615579, 627733	g7621	RPGP2	Rap1 GTPase−activating protein 2
LG11	641312	g7622	DLG5	Disks large homolog 5
LG11	654405, 654624	g7623	SPZ4	Protein spaetzle 4
LG11	810894	g7628	YELL	Protein yellow
LG11	897484	g7635	PAK1	Serine/threonine−protein kinase PAK 1
LG11	932237, 933730, 933917	g7637	ZN683	Tissue-resident T−cell transcription regulator protein ZNF683
LG11	993990, 994136, 994283, 996494	g7641	ESTF	Esterase FE4
LG11	1135366, 1135841	g7648	RYK	Tyrosine-protein kinase RYK
LG11	1157095	g7649	RYK	Tyrosine-protein kinase RYK

**Table 3 animals-14-00862-t003:** The emergence rate of the honeybees after injection.

Treatment	Number of Injections	Number of Pupae Emerged	χ^2^ Value	*p* Value
SiYell	78	58 (74.4%)	0.395	0.821
NC	53	40 (75.5%)		
Water	57	45 (78.9%)		
The comparison of different groups			χ^2^ value	*p* value
SiYell vs. NC			0.021	0.886
SiYell vs. Water			0.383	0.536
NC vs. Water			0.189	0.664

**Table 4 animals-14-00862-t004:** The color variants of the honeybees after injection.

Treatment	Number of Pupae Emerged	Number of the Adults with Color Variation	χ^2^ Value	*p* Value
SiYell	58	45 (77.6%)	58.838	0.000 **
NC	40	12 (30.0%)		
Water	45	2 (4.4%)		
The comparison of different groups			χ^2^ value	*p* value
SiYell vs. NC			22.030	0.000 **
SiYell vs. Water			54.642	0.000 **
NC vs. Water			10.053	0.002 **

** *p* < 0.01.

## Data Availability

Data are contained within the article and Appendix A.

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
