# Peer review of "Whole-Genome Resequencing−Based Qualitative Trait Locus Mapping Correlated yellow with the Mutant Color in Honeybees, Apis cerana cerana"

_animals, 2024, doi:10.3390/ani14060862_

Round 1
Reviewer 1 Report
Comments and Suggestions for Authors
Shao et. al., have efficiently and elaborately written the manuscript, “Whole-genome resequencing-based quantitative trait locus 2 mapping correlated yellow with the mutant color in honeybees, 3 Apis cerana cerana”
The authors have covered in-depth knowledge of coloration in terms of molecular biology, genomics, breeding, etc. The phenotypical difference was investigated by sequencing, and the hypothesis was confirmed using a functional study using RNAi. I believed that this research was properly designed and performed to derive the expected results. Great works. I have some minor comments to improve the manuscript.
Overall: check whether Honeybee or Honey bee are correct.
Line 33 : Drosophila , silkworms , beetles, stick insects à Drosophila is the only scientific name
Line 50: CRISPR/Cas9
Line 72: Ac maybe better to spread the word here. A. cerana
Line 74: a queen limiter what is this?
Line 75: use more scientific words than “cultivated naturally”
Line 174: space between number and ℃
Line 93: BLASTp
Line 183: 72 h space
Line 145: primer sets
Line 216: (optional) 1,177,784
Line 21: Check out whether small interfering RNA or short interfering RNA
Reviewer 2 Report
Comments and Suggestions for Authors
The authors clearly show that a naturally occurring frameshift mutation in the yellow gene is responsible for the br mutation in Apis cerana. I could not find any critical deficiency in the manuscript. Minor suggestions are listed bellow.
Figure 1A The wings are so shiny that it's hard to see the abdominal pattern, as well as the background is distracting. It is preferable to show not only dorsal but also ventral view.
Figure 1B is somewhat misleading to suggest that the genetic mappings shown in Fig. 1C and D were peformed using these drones.
Figure 3 is not necessary. Both g7623 and g7635 are not so closely located with g7628.
Figure 4A-C I could not undestand the purpose of arraying three similar sets of bees. Is there any special basis for selecting bees in each panel? I think that it is better to show as many abdominal enlarged views as possible for each treatment.
Reviewer 3 Report
Comments and Suggestions for Authors
The manuscript reports an interesting study focused on cuticle color in Apis cerana.
A few suggestions below can be addressed to improve the content:
1) Title - from the described content, it seems that the color mutant allele follows a Mendelian inheritance pattern - therefore it not clear the reference to a quantitative locus.
2) Introduction: it might better describe the question of the quantitative traits related to colors in honey bees - more information on what is also nown in Apis mellifera should be mentioned.
3) M&M: the title of paragraph 2.2 is not correct considering the content of the paragraph - Paragraph 2.1 should better descibe the honey bee populations investigated for the different activities
Paragraph 2.5 reports a methodology that is not very common in this contet: it is not clear what are the thresholds included in this analysis - how SNPs were included in this context and if genomic windows were considered. Usually Fst analyses are more informatives in these context
4) Conclusions are not reported
5) References should be checked and formatted according to the rules of the journal
Comments on the Quality of English LanguageEnglish is in general fine. It should be improved in a few sentences.
Reviewer 4 Report
Comments and Suggestions for Authors
Brief Summary:
This paper examined the genetic basis of a mutation in cuticle color in the honeybee Apis cerana cerana using genome resequencing of wildtype and mutant drones produced by a single virgin queen. A candidate locus was identified by calculating Euclidean distance between mutants and wildtypes at each SNP, performing Loess regression to fit a curve to these data, and setting a threshold of the top 0.5% Euclidean distance for candidate region selection. From this, genes with synonymous substitutions became candidate genes. One of these genes, the yellow gene, had a 2bp deletion causing a frameshift mutation. RT-QPCR of this gene as well as two neighboring genes was performed on RNA extracted from mutant and wildtype drones; gene expression was only significantly different between wild-types and mutants at the yellow gene. Finally, RNAi silencing of the yellow gene was used to reduce yellow gene expression in workers and putatively result in a lighter coloration.
Overall, this is an interesting study which provides new information on color patterns in insects. The identification of the gene involved is certainly notable. And the RNAi experimental work, which attempts to demonstrate that the gene in question is indeed responsible for the phenotypic variation, is admirable. Thus we view the study favorably overall. However, we had some important concerns about the work.
Major Comments:
● One of the key findings of this paper is that there was a mutation within the yellow gene, and that individuals with this mutant showed decreased expression of the gene. However, the link between a mutation in the 4th exon of the gene and decreased expression is unclear. Why would a mutation within the coding sequence cause a change in the expression? This must be clarified and explained.
● The success of RNAi was determined by simply looking at the emerged workers. But were researchers making these observations blind to treatment? The authors need to use a quantitative technique to document that there really is a difference in color and this isn’t just observational bias. The images on figure 4, suggest that the effect is quite subtle, if it is there at all.
Minor Comments:
● “Lowess regression” is mostly used throughout (in abstract: pg 1, line 17; in results: pg 5, line 210; in figure 1 caption: pg 6, line 221/222; in discussion: pg 9, line 281), but “Loess regression” is used in methods (pg 3, line 109). Keep consistent.
● Why only 6 WT and 6 mutant drones used for RT-QPCR of g7635 ? (And not full 11 WT and 10 mutant)? (pg 4, line 146)
● State in caption for figure 1C that colors correspond to chromosome # (I assume they do, but no explanation for color variation is given).
● The NC and Water treatments need to be be compared for color variation differences as indicated in table 3 (pg8). SiYell is compared for color variation to both NC and to water, and both were significantly different. And, in above table, all 3 comparisons were made for emergence rate (including NC vs. water). So we need to see the NC vs Water comparison as well, as it is informative.
Comments on the Quality of English LanguageN/A
